# Motivations for Social Withdrawal, Mental Health, and Well-Being in Emerging Adulthood: A Person-Oriented Approach

**DOI:** 10.3390/bs13120977

**Published:** 2023-11-27

**Authors:** Stefania Sette, Giulia Pecora, Fiorenzo Laghi, Robert J. Coplan

**Affiliations:** 1Department of Developmental and Social Psychology, Sapienza University of Rome, 00185 Rome, Italy; giulia.pecora@uniroma1.it (G.P.); fiorenzo.laghi@uniroma1.it (F.L.); 2Department of Psychology, Carleton University, Ottawa, ON K1S 5B6, Canada; robertcoplan@cunet.carleton.ca

**Keywords:** social withdrawal motivations, internalizing difficulties, latent profile analysis (LPA), emerging adulthood

## Abstract

Emerging adults seek solitude because of different underlying motivational and emotional processes. The current short-term longitudinal study aimed to: (1) identify subgroups of socially withdrawn emerging adults characterized by different motivations for solitude (shyness, unsociability, social avoidance) and affect (positive, negative); and (2) compare these subgroups in terms of indices of internalizing difficulties and life-satisfaction. Participants were *N* = 348 university students (*M*_age_ = 21.85 years, *SD* = 3.84) from Italy, who completed online questionnaires at two-time points separated by three months. Results from a latent profile analysis (LPA) suggested three distinct subgroups characterized by different social withdrawal motivations (i.e., shy, unsociable, and socially avoidant), as well as a non-withdrawn subgroup (characterized by low social withdrawal motivations, low negative affect, and high positive affect). Among the results, the socially avoidant subgroup reported the highest levels of social anxiety, whereas the avoidant and shy subgroups reported the highest loneliness and lowest life satisfaction. The unsociable subgroup appeared to be the most well-adjusted subgroup of socially withdrawn emerging adults and reported similar levels of life satisfaction as the non-withdrawn subgroup. Our findings confirmed the heterogeneity of emerging adults’ experiences of solitude, with different motivations for social withdrawal appearing to confer a differential risk for maladjustment.

## 1. Introduction

Emerging adulthood is a period of life characterized by substantive changes in many developmental domains involving biological, cognitive, and emotional processes [1]. During this critical developmental phase, solitary experiences acquire new meanings and functions for young people. On the one hand, the time emerging adults spend alone can provide valuable opportunities for developing autonomy and establishing a differentiated identity [2,3,4], which represent fundamental tasks for this specific life phase. Time alone at this age can also represent a healthy coping strategy for facing social pressure and other demanding challenges, getting back in touch with personal interests, and regulating behaviors [5,6]. On the other hand, when emerging adults spend too much time alone—or when solitude becomes a mechanism to avoid social interactions experienced as anxiety-proving or unpleasant—it can also represent a risk factor for the onset of socialization difficulties, internalizing problems, and occupational issues [7,8]. According to Baumeister and Leary’s [9] theoretical model (see also [10]), humans are driven by the *need to belong*, which propels them to form and maintain interpersonal relationships, which, in turn, promote positive affect and well-being. However, when belongingness needs are unmet, it can lead to negative affective outcomes, including feelings of anxiety, depression, and/or loneliness.

In this study, we focused our attention on aspects of solitude related to *social withdrawal*, which is defined as the tendency to remove oneself from opportunities for social interactions with familiar and unfamiliar peers [11]. Socially withdrawn individuals tend to avoid social situations and spend more time alone than their more sociable counterparts [12]. However, social withdrawal is a heterogeneous and multidimensional construct, with distinct motivations for seeking solitude associated with different consequences, including increased risk for depression, social anxiety, and feelings of loneliness [13,14].

There is growing recent interest in the links between social withdrawal and indices of well-being among emerging adults [5,15,16]. However, only a few studies have employed a *person-oriented* approach [13,15,17], which focuses on potential differences among groups of individuals (employing statistical techniques such as profile or class analyses [18]). This differs from a *variable-oriented* approach, which describes associations among variables among all group members (using statistical techniques such as correlations or regressions [18]). More specifically, variable-oriented approaches typically assume a homogeneous population, implying that associations between variables are uniform across all individuals. This may obscure significant individual distinctions [18]. Importantly, emerging adulthood is a unique period characterized by significant developmental changes [1]. From this perspective, a person-oriented approach can help disentangle the intricate connections between various variables by uncovering the underlying structure of the data and providing insights into how social withdrawal motivations might differentiate subgroups of emerging adults. Accordingly, we employed a person-oriented approach in order to: (1) identify subgroups of socially withdrawn emerging adults characterized by different motivations for solitude (shyness, unsociability, and social avoidance) and affect (positive and negative); and (2) compare these groups in terms of their association with indices of internalizing problems (i.e., depression, social anxiety, and loneliness) and life satisfaction—both concurrently and three-months later. 

### 1.1. Motivations for Social Withdrawal

Asendorpf’s [19] classic theoretical model characterized social withdrawal as a multidimensional construct defined by the interplay between social approach (i.e., the desire to interact with others) and avoidance (i.e., the tendency to refrain from social interactions and avoid others) motivations. Contemporary perspectives now conceptualize social withdrawal using more complex theoretical mechanisms to identify distinct subtypes of social withdrawal characterized by different underlying motivational and affective substrates [20,21]. A similar theoretical framework has been proposed by Elliot et al. [22] (see also [23]), who postulated the presence of approach and avoidance social goals that drive individuals toward positive outcomes of social relationships (e.g., deepening one’s relationships or being accepted) or away from negative outcomes of social relationships for fear of receiving negative evaluations from others (e.g., avoiding conflict in one’s relationships or rejection). In this model, social avoidance motivations are not considered as the avoidance of social interactions per se but the avoidance of threats in social situations. Overall, avoidance motivations are generally associated with negative affect (e.g., nervousness), whereas approach motivations are related to positive affect (e.g., happiness).

The prevailing conceptual models guiding contemporary research on social withdrawal identify and describe three distinct subtypes of social withdrawal: shyness, unsociability, and social avoidance [11,19,22]. *Shyness* characterizes individuals with ambivalent dispositions to interact with others and seek solitude simultaneously, reflecting an internal conflict between high social approach and avoidance motivations [11]. The term shyness evidences conceptual overlap with other related constructs, such as behavioral inhibition, social reticence, and anxious solitude [24,25]). However, these terms share an underlying theme of wariness in social situations. For the current manuscript, we consider these terms functionally equivalent. 

Shy emerging adults often experience high levels of discomfort and anxiety in social situations (particularly among unfamiliar peers), so they may spend much time alone despite a desire to interact with others [13]. As a result, shy individuals may miss out on opportunities to develop social and emotional skills, which can exacerbate clinical social anxiety and adjustment problems [14]. Peer relationships are particularly central during emerging adulthood, since youths spend more time with friends and less time with parents. For instance, emerging adults face unique challenges during the transition to the university environment, such as transferring from senior high school to university and creating new supportive social relationships [8]. Shy emerging adults, who tend to display social anxiety in new contexts, also report more difficulties building new supportive friendships, and their existing relationships are more likely to be characterized by less affection and confidence [26,27]. For instance, Nelson et al. [27] found that U.S. emerging adults attending university with higher levels of shyness reported lower relationship quality with friends, parents, and romantic partners. Similarly, McVarnock and Closson [16] reported that shyness was positively related to test anxiety (i.e., having upset feelings during tests) and negatively associated with intrinsic values (i.e., liking what is being taught in courses) and self-efficacy (i.e., learning the materials for courses) in a sample of Canadian university student emerging adults. However, not all shy individuals are at risk for poorer adjustment. For example, it has been suggested that the positive affect experienced during social situations may yield a more positive type of shyness, allowing these individuals to regulate their arousal and engagement with others [28]. 

The second subtype of social withdrawal is *unsociability*, which reflects low approach and avoidance motivations to interact with others [11,19]. Unsociability also evidences conceptual overlap with other related constructs, such as affinity for aloneness, preference for solitude, and social disinterest [29]. However, all these terms share a common underlying theme of a non-fearful preference for (enjoyment of) solitude—and we consider them functionally equivalent for this study. Unsociable emerging adults also tend to spend less time in social situations, but unlike their shy counterparts, they do not feel fearful or anxious while interacting with peers. Instead, they spend more time alone because they value and enjoy solitude [11]. It has been widely suggested that unsociability is a comparatively benign form of social withdrawal, as it does not tend to be associated with internalizing difficulties and relationship problems [14,15,30]. For instance, McVarnock and Closson [16] did not find significant associations between unsociability, self-efficacy, and test anxiety in a sample of university students. 

The last subtype of social withdrawal is *social avoidance*, characterized by a low approach motivation and high avoidance motivation toward social interactions. This subtype of social withdrawal has been the least studied empirically [11]. Avoidant emerging adults are believed to be not only disinterested in social interactions (in contrast to their shy counterparts), but also prone to actively avoiding opportunities for peer interaction (in contrast to their unsociable counterparts) [11]. Some researchers have proposed that social avoidance might derive from high levels of social anxiety felt during exposure to negative peer experiences, such as rejection or victimization [31]. Others have suggested that this socially withdrawn subtype may represent a highly severe form of shyness [11]. In this regard, socially avoidant individuals would experience such high social anxiety, discomfort, and fear of social judgments that their desire to approach others would progressively decrease until complete extinction. Previous studies have also reported links between social avoidance and social anhedonia (i.e., the lack of pleasure from social interactions [8,32]). For example, Bowker et al. [32] found that although both shyness and social avoidance were related to social anhedonia in a sample of emerging adults in the U.S., the association between social avoidance and social anhedonia was stronger than the link between shyness and social anhedonia. 

### 1.2. Implications of Motivations for Social Withdrawal in Emerging Adulthood

There is now considerable research indicating that different subtypes of social withdrawal are differentially associated with indices of socio-emotional functioning, mental health, and well-being [11]. However, most of this research has been conducted in childhood and adolescent samples—and less is known about the implications of social withdrawal subtypes in emerging adults. Emerging adults face unique challenges, such as entering the university environment and creating new supportive social relationships [8]. Compared to their more sociable counterparts, socially withdrawn emerging adults report more difficulties building new supportive friendships and are at greater risk of developing internalizing difficulties [26,27]. Therefore, examining the implications of distinct forms of social withdrawal in this unique developmental period is particularly relevant. 

The implications of shyness have been the most studied. Shyness is a relatively stable temperamental trait throughout development [11]. It is also robustly related to internalizing problems (e.g., negative affect, social anxiety, loneliness) and peer difficulties (e.g., rejection, victimization) from early childhood through emerging adulthood [31,33]. Among emerging adults, shyness has demonstrated associations with depressive symptoms, existential concerns, test anxiety, and lower levels of academic self-efficacy [16,34]. Indeed, Closson et al. [35] found that shy young adults reported the lowest levels of social support, happiness, life satisfaction, and self-worth compared to other groups of withdrawn and non-withdrawn peers. 

There has been growing interest in the implications of unsociability [36], which is generally linked to a lower risk of social maladjustment [37,38]. As aforementioned, unsociability represents a non-fearful preference for solitude rather than a negative attitude determined by emotional or social concerns. Coplan et al. [36] discussed the potential of *developmental timing* effects related to unsociability, arguing that the pattern of associations between unsociability and socio-emotional problems increases from early childhood to early adolescence—and then decreases from early adolescence to emerging adulthood. In support of these assertions, unsociability appears to be strongly associated with peer problems in late childhood and adolescence, when expectations for peer interaction are high, along with increased peer pressure and desire for conformity, leading to solitary behaviors being viewed as quite deviant [11]. Conversely, during adolescence, preference for solitude becomes progressively more adaptive [39], as attitudes toward spending time alone become more positive and normative [40]. This positive trend regarding unsociability appears to continue into emerging adulthood [15]. For example, Nelson et al. [41] found that unsociability among emerging adults was not related to problematic media use, depression, or externalizing problems one year later. Similarly, Bowker et al. [32] reported that unsociability among emerging adults was not related to anhedonia or anxiety sensitivity but was positively associated with creativity. 

Finally, social avoidance is a subtype of social withdrawal that appears to have the most adverse effects on adjustment [11]. Indeed, as compared to their shy and unsociable counterparts, socially avoidant children and young adolescents reported the most pervasive socio-emotional problems, with the lowest levels of positive affect and the highest levels of negative affect [13]. Results from previous studies also indicate that avoidant adolescents and emerging adults experience a range of socio-emotional difficulties, including poorer quality relationships, and lower self-esteem, as well as higher levels of emotion dysregulation, social anhedonia, and depression [14,32,42]. In a sample of undergraduate students, Nelson et al. [14] found that the socially avoidant group reported higher scores on suicidal ideations and self-harm and lower scores on paternal relationship quality than the shy and unsociable groups. Most recently, Bowker et al. [7] conducted a 10-country cross-cultural study of social withdrawal among university students and reported that social avoidance was positively associated with loneliness in all sites. Examining the motivations underlying adolescents’ social withdrawal can provide helpful insights into the processes that lead adolescents and emerging adults to forgo social opportunities and shed light on the possible consequences of socio-emotional adjustment. 

### 1.3. The Present Study 

The present study aimed to identify subgroups of emerging adults based on their social withdrawal motivations and positive/negative affect. To accomplish this purpose, we adopted a person-oriented approach and employed latent profile analysis in a longitudinal sample of emerging adults. In particular, we first aimed to delineate different subgroups characterizing emerging adults’ tendency to spend time alone based on individual differences in social motivations (i.e., shyness, unsociability, social avoidance) and affect (i.e., positive, negative). Accordingly, university students completed assessments at two time points near the beginning (Time 1) and at the end of a university term (Time 2, 3-months later; for a study aimed to assess students’ adjustment using a similar time interval see [43]). At Time 1, we expected socially withdrawn emerging adults to differentiate themselves into three subgroups: shy, unsociable, and socially avoidant [13]. We also speculated to identify a subgroup characterized by lower levels of social withdrawal motivations and higher positive affect, representing more sociable individuals. 

We also investigated how the individuated profiles at Time 1 differed in terms of indices of internalizing problems and life satisfaction, both concurrently and at Time 2 (3-months later, to reflect the start and end of classes). It was hypothesized that these social withdrawal subgroups would differ regarding maladaptive developmental outcomes, both contemporaneously and three months later. Specifically, we expected groups characterized by shyness, social avoidance, and negative affect to report higher depression, social anxiety, and feelings of loneliness, as well as lower life satisfaction, compared to those who displayed more unsociability [14,32,35,43]. 

Finally, it is important to consider gender differences in the meaning and implications of social withdrawal, but to date, the empirical evidence in this regard remains mixed [35,44]. For instance, Closson et al. [35] did not report significant gender differences in socially withdrawn emerging adults’ well-being (e.g., social support, happiness, life satisfaction), whereas Nelson et al. [44] found that socially avoidant females reported lower self-image than males in a sample of established adults (aged 30–35 years). Accordingly, gender effects were tested on a more exploratory basis.

Previous studies of motivations for social withdrawal among emerging adults that have adopted a person-oriented approach remain scarce [13,15,17], and none have included a specific measure of social avoidance. In this regard, the present study seeks to fill a gap in the extant literature and expand our knowledge of the implications of social withdrawal among emerging adults across two time-points spanning a university semester. 

## 2. Methods

### 2.1. Study Design

The study employed a longitudinal design with data collection at two time points (approximately three months apart). For this study, we recruited a convenience sample of university students. 

### 2.2. Setting

After providing informed consent, participating university students completed assessments of motivations for social withdrawal, affect (positive, negative), internalizing problems (i.e., depression, social anxiety, loneliness), and life satisfaction. Measures were completed online via Limesurvey at T1 (March 2017) and T2 (approximately three months apart). 

### 2.3. Participants

Participants were *N* = 348 university students (90% females; *M*_age_ = 21.85, *SD* = 3.84; *Time 1*, T1) enrolled in a psychology course in a large urban center in Italy. Of these, *n* = 159 students (97% females) participated again approximately three months later, at the end of classes (*Time 2*, T2). A large majority of the sample (96%, *n* = 333) self-identified as Caucasian, with a variety of other ethnicities also represented (2.9% Hispanic/Latin, 0.6% Asiatic, 0.3% Arabic, and 0.6% “Other”).

### 2.4. Measurement

*Social Withdrawal Motivations*. Motivations for social withdrawal were assessed using the *Social Preference Scale—Revised* (SPS—R; [7]). The scale was translated into Italian and then back-translated into English by independent blinded experts. A multiple-group factor analysis alignment identified the items that displayed the best psychometric properties across countries, leading to the final version including 10 items out of 21 of the original form. SPS-R is a self-report questionnaire assessing social withdrawal motivations during adolescence and emerging adulthood. Participants answered each item on a 5-point scale (from 1 = *not at all*, to 5 = *a lot*). The measure is composed of three subscales. In the current study, the shyness subscale included 4 items (e.g., “Feeling nervous to interact with others despite the desire to do so”) and showed good reliability at both T1 (*α* = 0.87) and T2 (*α* = 0.83). The unsociability subscale included 2 items (e.g., “Preferring to hang out with others than to spend time alone”—reverse-scored) and showed acceptable internal consistency at both T1 (*α* = 0.76) and T2 (*α* = 0.66). Finally, the avoidance subscale, including 4 items (e.g., “Choosing to spend time alone due to dislike of others”), was found to be reliable at both T1 (*α* = 0.84) and T2 (*α* = 0.82). 

*Positive and negative affect*. Affect was assessed using the *Positive and Negative Affect Schedule* (PANAS; [45]; for use in Italy see [46]), which is commonly used to measure mood and emotions. This scale includes 20 items, with 10 items examining positive affect (e.g., excited, determined) and 10 examining negative affect (e.g., nervous, distressed). Participants rated the extent to which they experienced a specific affect on a 5-point scale (from 1 = *very slightly or not at all*, to 5 = *extremely*). Reliability was good for both scales and time points (positive affect: *α*_t1_ = 0.81; *α*_t2_ = 0.85; negative affect: *α*_t1_ = 0.83; *α*_t2_ = 0.89). 

*Loneliness*. Loneliness was assessed via the 20-item version of the *UCLA Loneliness Scale* ([47]; for use of the scale in Italy, see [48]). The scale was used to investigate subjective feelings of loneliness and social isolation. For this study, consistent with Bowker et al. [7], we considered 5 items (e.g., “I feel isolated from others”), which were found to be appropriate for cross-cultural comparisons [49]. Participants rated each item on a 4-point scale from 0 (“*I never feel this way*”) to 3 (“*I often feel this way*”). Reliability for this measure’s short version was good at T1 (*α* = 0.84) and T2 (*α* = 0.82).

*Social anxiety*. Participants completed the *Social Interaction Anxiety Scale* (SIAS; [50]; for use in Italy see [51]), a widely used measure of social anxiety in both clinical and non-clinical populations. The instrument includes 19 items (e.g., “I have difficulty making eye contact with others”) on a 5-point scale, from 0 (*not at all characteristic or true of me*) to 4 (*extremely characteristic or true of me*). Reliability was high at T1 (*α* = 0.94) and T2 (*α* = 0.93).

*Depression*. Participants also completed the Italian version of the *Beck Depression Inventory* ([52]; for its use in Italy see [53]), a widely used self-report questionnaire to screen for depressive symptoms in adolescents and adults. It is comprised of 21 items rated on a 4-point scale, indicating progressively greater severity symptoms (e.g., “I have as much energy as ever”; “I have less energy than I used to have”; “I don’t have enough energy to do very much”; “I don’t have enough energy to do anything”, coded as 0, 1, 2, and 3, respectively). The item evaluating suicidal ideation was removed for this study, thus reducing the total number of items to 20. Internal reliability was good at T1 (*α* = 0.87) and T2 (*α* = 0.92).

*Life Satisfaction*. Finally, participants completed the *Satisfaction With Life Scale* (SWLS; [54]; for its use in Italy see [55]) to examine global cognitive judgments of life satisfaction. The scale consists of 5 items (e.g., “In most ways my life is close to my ideal”), which are statements that participants have to rate using a 7-point scale, from 1 (*strongly disagree*) to 7 (*strongly agree*). Reliability for this measure was good at T1 (*α* = 0.85) and T2 (*α* = 0.86).

### 2.5. Statistical Methods

We first computed descriptive statistics and Pearson’s correlations among the study variables. Based on our theoretical framework (e.g., [20,21] and for more direct comparisons with previous studies [13]), we conducted a latent profile analysis (LPA) to individuate the number of emerging adulthood subgroups considering positive affect, negative affect, shyness, unsociability, and social avoidance at T1 (*N* = 348 university students). We estimated models with one- to sixth-class solutions with 20 random sets of starting values in the first stage and 5 optimizations in the last stage (we did not include any other model specifications). To better interpret each class solution, each variable at T1 was first standardized in the LPA. We used multiple fit indices, model parsimony, and theory to choose the best class solution. In terms of model fit indices, for the best class solution, we considered (1) the lower log-likelihood, Akaike information criterion (AIC), the Bayesian information criterion (BIC), and the sample-adjusted BIC, (2) an entropy closer to 1, and (3) a significant Lo–Mendell–Ruben (LMR) and bootstrap likelihood ratio test (BLRT; [56]). Also, for the best class solution, each individuated subgroup had to represent at least 5% of the overall sample ([57]). We estimated models using the maximum likelihood estimator with robust standard errors (MLR). Then, to understand possible differences between the individuated subgroups on variables at T1 included in the LPA and other additional variables, both at the baseline and three-months later (i.e., social withdrawal motivations at T2, internalizing difficulties at T1 and T2, and satisfaction with life at T1 and T2), we conducted a series of analyses of variance (ANOVAs), with Bonferroni post hoc comparisons. At T1, we considered the total sample of participants (*N* = 348 university students); at T2, we only considered university students who also completed T2 assessments (*n* = 159). SPSS 27 and MPlus 8.4 [58] were used to run the analyses. 

## 3. Results

### 3.1. Preliminary Analyses

Descriptive statistics and correlations among the study variables are reported in Table 1 and Table 2. Overall, the pattern of correlations among the study variables was in the expected direction. For instance, shyness at T1 was positively related to high social anxiety, depression, and loneliness at T1 and T2. Unsociability at T1 was positively correlated with social anxiety and loneliness at T1 but not at T2. Finally, social avoidance at T1 was positively associated with social anxiety, depression, and loneliness both at T1 and T2. Males reported higher scores on T1 social avoidance and T2 negative affect than females.

### 3.2. Latent Profile Analysis (LPA): Social Withdrawal Motivations and Affect at T1

We estimated sixth-class solutions with the total sample at T1 (*N* = 348), as reported in Table 3. Model 4 reported the best model fit for the data compared to all the tested LPA models. The model consisted of four subgroups, which were labeled, in line with the previous literature, shy (*n* = 59, 16.9%), unsociable (*n* = 41, 11.8%), socially avoidant (*n* = 23, 6.6%), and non-withdrawn (*n* = 225, 64.7%). The results for each subgroup are displayed in Figure 1. Overall, the results from a series of ANOVAs, with Bonferroni post hoc comparisons, revealed that the shy subgroup reported higher scores on shyness than their unsociable and non-withdrawn counterparts (but lower than the socially avoidant subgroup), *F*(3, 344) = 429.54, *p* < 0.001, *η^2^_p_
*= 0.79. The unsociable subgroup reported higher scores of unsociability than the shy and non-withdrawn subgroups (but no differences emerged between the unsociable and socially avoidant subgroups), *F*(3, 344) = 36.41, *p* < 0.001, *η^2^_p_
*= 0.24. The socially avoidant subgroup displayed the highest scores on social avoidance, *F*(3, 344) = 259.37, *p* < 0.001, *η^2^_p_
*= 0.69. Finally, the non-withdrawn subgroup reported higher scores of positive affect, *F*(3, 344) = 15.51, *p* < 0.001, *η^2^_p_
*= 0.12, and lower scores of negative affect, *F*(3, 344) = 23.85, *p* < 0.001, *η^2^_p_
*= 0.17, than the other socially withdrawn subgroups, which did not differ among them. 

No significant age differences emerged between the latent profile subgroups, *F*(3, 344) = 1.56, *p* = 0.20, *η^2^_p_
*= 0.01. However, the results from Fisher’s exact test indicated significant gender differences (*p* = 0.01). The results from follow-up tests (standardized residuals) indicated that the unsociable subgroup was composed of a higher number of males than expected (*n* = 10 males, 24%, vs. *n* = 31 females, 76%). No other differences between the observed and the expected frequencies were observed in the shy (*n* = 10 males, 17%, vs. *n* = 49 females, 83%), socially avoidant (*n* = 1 males, 4%, vs. *n* = 22 females, 96%), and sociable (*n* = 15 males, 7%, vs. *n* = 210 females, 93%) subgroups.

### 3.3. Characteristics of Social Withdrawn Subgroups at T1 and T2

To compare the four individuated latent profile subgroups, we conducted several ANOVAs on the other study variables at T1 (*N* = 348 university students) and T2 (3-months later; *n* = 159 university students who also participated at this time point; Table 4). We report the results without controlling for gender because each subgroup was composed of a small number of males compared to females. However, when this control was included, the results of the ANOVAs were equivalent to those reported in Section 3, except for T1 life satisfaction and T2 depression. For T1 life satisfaction, the results did not reveal differences between the non-withdrawn and the socially avoidant subgroups. For T2 depression, the findings suggested that the shy subgroup reported higher scores than the non-withdrawn subgroup. At T2, the Bonferroni post hoc comparisons did not consider the socially avoidant subgroup since it comprised only a few participants (*n* = 3). The findings revealed that the *shy* subgroup displayed higher social anxiety and loneliness scores at T1 than the unsociable and non-withdrawn subgroups. At T2, the shy subgroup reported higher social anxiety and loneliness scores than the non-withdrawn subgroup (no differences emerged with the unsociable subgroup at T2). The shy subgroup reported higher scores on shyness at T2 than the unsociable and non-withdrawn subgroups and lower scores on positive affect than the non-withdrawn subgroup. The *unsociable* subgroup displayed higher scores on loneliness and social anxiety at T1 than the non-withdrawn subgroup. No significant differences emerged between the unsociable and non-withdrawn subgroups on life satisfaction at T1. At T2, the unsociable subgroup reported higher scores on social anxiety than the non-withdrawn subgroup. The unsociable subgroup reported higher scores on unsociability at T2 than the non-withdrawn subgroup and higher scores on avoidance than the shy and non-withdrawn subgroups. No differences emerged on loneliness, depression, and life satisfaction at T2 between the unsociable and non-withdrawn subgroups. 

The *socially avoidant* subgroup displayed higher levels of social anxiety at T1 than the shy, unsociable, and non-withdrawn subgroups. The socially avoidant subgroup did not differ from the shy subgroup for loneliness and life satisfaction at T1. However, the socially avoidant subgroup reported higher scores on loneliness and lower scores on life satisfaction at T1 than the unsociable and non-withdrawn subgroups. The *non-withdrawn* subgroup displayed lower levels of depression at T1 than the shy, unsociable, and socially avoidant subgroups, which did not differ among them.

## 4. Discussion 

Investigating emerging adults’ motivations for solitude is crucial to understanding the processes underlying social withdrawal and its potential associations with socio-emotional functioning [8,15]. Some previous research has examined how different social withdrawal motivations are associated with indices of internalizing problems and well-being among emerging adults [32,35,42]. However, few studies have taken a person-oriented approach (e.g., [13]). This study identified subgroups of emerging adults based on their social withdrawal motivations (i.e., shyness, unsociability, social avoidance) and affect (i.e., positive, negative). We also investigated whether such subgroups differed in terms of indices of maladjustment (i.e., social anxiety, depression, feelings of loneliness, and life dissatisfaction) concurrently and three-months later (to reflect the start and end of a semester of university). In line with theoretical models and empirical studies of social withdrawal [19,22], our results individuated three socially withdrawn profiles characterized as shy, unsociable, and socially avoidant. A fourth subgroup was comprised of non-withdrawn emerging adults. These findings support prior research individuating different subtypes of social withdrawal (e.g., [13,17]). Moreover, we found that emerging adults with higher levels of social avoidance and shyness reported higher internalizing problems (i.e., social anxiety, loneliness) and dissatisfaction with life compared to their unsociable and non-withdrawn counterparts. Overall, our results indicated that investigating social withdrawal motivations is crucial for understanding how the tendency to spend time alone can play a role in fostering pervasive associations with maladjustment during emerging adulthood. 

### 4.1. Subgroups of Socially Withdrawn Emerging Adults

Results from the latent-profile analysis explored individual differences in our sample by delineating four emerging adult subgroups, differing with regard to social withdrawal motivations and affect. The largest subgroup (which included almost 65% of the sample) described *non-withdrawn* individuals, who were characterized by the lowest levels of shyness, unsociability, and social avoidance, as well as the highest level of positive affect and lowest negative affect, compared to all other subgroups. This result supports prior studies’ findings that non-withdrawn individuals are generally characterized by adequate emotional well-being and do not present difficulties in social situations [8,13].

The second largest subgroup (about 17% of the sample) was characterized by higher levels of shyness than the unsociable and non-withdrawn subgroups, as well as lower levels of avoidance and unsociability compared to the other two socially withdrawn subgroups. We labeled this socially withdrawn subgroup as *shy* since it included emerging adults with an internal conflict between the desire to interact with others (i.e., low unsociability and social avoidance) and a fearful approach in social situations (i.e., high shyness and negative affect; [8,11]). In other words, although shy emerging adults might be motivated to approach others and engage socially, their desires are inhibited by social anxiety and socio-evaluative concerns [14]. 

The third subgroup (about 13% of the sample) was labeled *unsociable*, characterized by comparatively high unsociability, moderate social avoidance, and low shyness (as compared to the other withdrawn subgroups). This group also reported moderate negative and positive affect compared to other subgroups. Finally, the smallest subgroup (about 7% of the sample) was labeled as *socially avoidant*, characterized by the highest levels of social avoidance, shyness, and unsociability, as well as high negative affect and low positive affect. In line with prior research [13,14], this subgroup appeared to be the most impaired compared to all other subgroups, with a strong tendency to avoid social situations, high negative emotionality, and low approach. Notably, the socially avoidant subgroup reported the highest mean levels of shyness, which supports the assumption that social avoidance represents a highly severe form of shyness (e.g., [11]). It has also been suggested that chronic experiences of peer rejection and high levels of fear and anxiety during social situations over time serve to “extinguish” socially avoidant individuals’ desire to approach others and exacerbate their avoidance motivations [8].

### 4.2. Implications of Socially Withdrawn Motivations

#### 4.2.1. Social Anxiety

All subgroups differed significantly regarding social anxiety at T1, with the socially avoidant subgroup reporting the highest means, followed by the shy, unsociable, and non-withdrawn subgroups, respectively. At T2, the socially avoidant subgroup was excluded from comparisons due to the small sample size. However, the results indicated that shy and unsociable emerging adults reported higher social anxiety than the non-withdrawn group. Our findings align with the existing literature, suggesting that individuals who strongly prefer solitude and avoid social interactions are at higher risk of developing clinical social anxiety [44]. The shy subgroup reported significantly lower means than their socially avoidant counterparts at T1. Indeed, although they may experience fear and discomfort in social situations, shy individuals intimately desire to interact with others. This internal motivation to approach others could serve to at least partially reduce the development of the most pervasive effects of social anxiety. Also, in line with previous studies [37,38], unsociability was the most benign form of social withdrawal, with the lowest means of social anxiety compared to the shy and socially avoidant subgroups. This result was likely due to the nature of this socially withdrawn motivation, representing a preference for solitude free of those fearful tendencies and social concerns that generally result in more severe socio-emotional difficulties [12]. 

#### 4.2.2. Depression

At T1, all three socially withdrawn subgroups reported higher symptoms of depression than the non-withdrawn subgroup but did not differ significantly from one another. These findings were partly in line with expectations. Although we speculated that the shy and avoidant subgroups would report higher levels of depressive symptoms [17], we also expected to find no significant differences between the unsociable and non-withdrawn subgroups. This expectation was based on previous research showing that unsociable adolescents and emerging adults reported lower levels of internalizing problems compared to their shy and avoidant counterparts [32,41]. However, the higher levels of negative affect (i.e., as high as in the shy and socially avoidant subgroups) could help to explain this association with depressive symptoms in the unsociable subgroup. 

Emerging adults face different challenges during their university years, such as transferring from senior high school to the university, moving to a new city, or initiating social interactions with novel peers [59]. The critical developmental period of our students may have exacerbated the levels of depression in all socially withdrawn subgroups, including the unsociable subgroup. At T2, no differences emerged among all subgroups, most likely because of the comparatively small sample size and the exclusion of the socially avoidant subgroup from comparisons. However, it should also be noted that, as we will discuss, the unsociable group did not differ from the non-withdrawn comparison group in terms of loneliness or life satisfaction.

#### 4.2.3. Loneliness

The results at T1 revealed that the socially avoidant and shy subgroups both reported higher means of loneliness than the unsociable and non-withdrawn subgroups. This finding was expected since shy and socially avoidant emerging adults experience negative affect and anxiety in social situations, which can enhance their tendency to spend time alone, thus leading to more intense feelings of loneliness (e.g., [31]). The results at T2 support this interpretation since the shy subgroup again showed significantly higher means of loneliness than the non-withdrawn group. In other words, when the desire to approach others is not fully satisfied, shy emerging adults may be dissatisfied with their social network [7]. Of note, the unsociable subgroup did not differ significantly from the non-withdrawn subgroup in terms of loneliness at either T1 or T2. This result further corroborates the hypothesis that a non-fearful proneness for solitude can be considered a relatively benign form of social withdrawal, which becomes more adaptive in this developmental stage [32,40,41].

#### 4.2.4. Life Satisfaction

Similar to the results for loneliness, the shy and socially avoidant subgroups reported lower life satisfaction than the unsociable and non-withdrawn subgroups. These findings were replicated at T2 (except for the socially avoidant subgroup, which was excluded from these analyses). Once again, individuals with the worst consequences at the expense of adjustment and well-being scored higher on negative affect and social anxiety and lower on positive affect. These findings are in line with other studies (e.g., [35]) and suggest that distancing oneself from social opportunities because it feels hard to face such situations has different meanings and more negative implications for adjustment compared to deciding to spend time alone because of a preference and enjoyment for solitary activities [8]. From this perspective, clarity on the motivations behind the choice to withdraw from others can disentangle processes that seem similar but are profoundly different.

### 4.3. Limitations, Strengths, and Future Directions

The present study focused on social withdrawal during emerging adulthood, with a novel approach that permitted the exploration of the individual differences characterizing social withdrawal motivations and investigating whether and how these differences were related to indices of maladjustment. We also followed up a sub-sample of students three months subsequent to the first data collection, providing short-term longitudinal data not typical of previous studies of social withdrawal among emerging adults. The focus on emerging adulthood is also relevant, since this represents a critical phase in which solitude and time spent alone not only heighten the risk for negative outcomes, but also offer a context beneficial to individual development (e.g., [15]) and which is remarkably adaptive for the achievement of specific developmental tasks (e.g., identity differentiation, autonomy, occupation obtainment; e.g., [5]). Despite these strengths, our study presents some limitations that should be considered for future research. 

First, there was a low participation rate at T2 (three-months after T1), especially for males, which led to the exclusion of the socially avoidant subgroup from analyses and might have weakened the power of our results. We did not assess the reasons for the participant attrition, but it is possible that the high avoidance characterizing this subgroup also affected their participation at T2. Similarly, the low proportion of males in our sample made it difficult to assess potential gender differences among the groups. Because the evidence is mixed on whether socially withdrawn emerging adult males and females differ in terms of indices of maladjustment/adjustment [35,44], it would be of interest for future research to test the moderating role of gender in these associations. It would also be relevant to consider how other individual factors (e.g., socio-economic status, sexual orientation, and ethnicity) intersect and influence experiences of solitude and indices of maladjustment (see [60] for a similar approach). Also, since this study adopted a quantitative approach with an online questionnaire, using qualitative methods (e.g., interviews and focus groups) could provide a deeper understanding of emerging adults’ motivations and experiences of solitude [6,61].

The three-month time frame from T1 to T2 was intended to reflect the start and end of a semester of university, but a larger temporal window between time points should be adopted in future studies to better understand the longer-term implications of different developmental trajectories of social withdrawal in emerging adulthood. In addition, future research could explore a wider range of adjustment outcomes across other domains relevant to emerging adults in university (e.g., academic success, romantic relationships, substance use, stress, etc.; e.g., [16]) Furthermore, we believe that the role of protective factors for social withdrawal during this specific developmental phase should be explored in future research, such as positivity, characterizing individuals who have the dispositional tendency to view themselves, their own life, and the future through a positive lens [62]. Future research should also aim to assess positive shyness, to see if such a group might emerge based on emerging adults’ ability to display positive emotional expressions and engage with others [28]. Finally, our study only focused on university students from Italy recruited using convenience sampling, which limits the generalizability of the results to other cultural contexts and to the population. Indeed, although there are some similarities in the consequences of social withdrawal motivations across cultures, there are also cultural-specific patterns [7]. Thus, to enhance the applicability of the study findings to other cultural contexts, it is essential to replicate this research across different cultural backgrounds.

## 5. Conclusions

In conclusion, a person-oriented approach allowed us to identify distinct groups of socially withdrawn emerging adults with different socio-emotional profiles. The results from our study are relevant not only to scholars but also provide helpful indications to practitioners (especially those in university settings), policymakers, parents, and emerging adults themselves for implementing intervention programs to reduce the costs of social withdrawal during the challenging period of university years, especially for the more at-risk subgroups (i.e., shy and socially avoidant). Therefore, intervention programs could increase practitioners’, policymakers’, and parents’ awareness about the different motivations and experiences of solitude and direct their attention to shy and socially avoidant individuals with high social anxiety, loneliness, and life dissatisfaction. Enhanced psychological support programs could be implemented for shy and socially avoidant emerging adults to address issues related to anxiety and depression during a period of life marked by unique challenges. For instance, specific training programs aiming at enhancing emotional and social skills could benefit shy and socially anxious people [63]. This also could allow practitioners, policymakers, and parents to not over-pathologize all experiences of solitude since some socially withdrawn individuals reported similar levels of life satisfaction as the non-withdrawn subgroup. 

## Figures and Tables

**Figure 1 behavsci-13-00977-f001:**
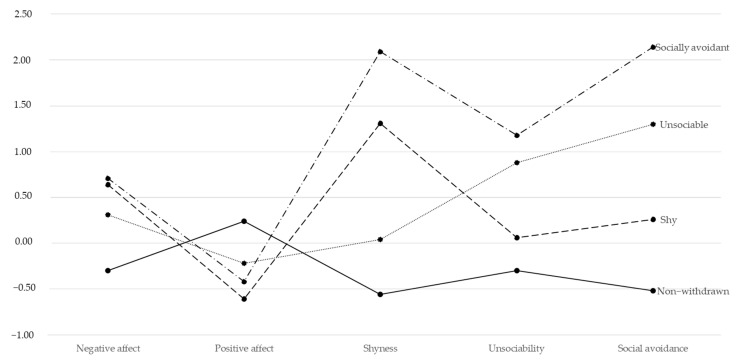
Results of latent profile analysis (LPA) with four class solution. Note: For each variable, we represent the standardized means.

**Table 1 behavsci-13-00977-t001:** Descriptive statistics for the study variables.

	*Mean*	*SD*	Range
*Time 1* (*N* = 348)			
Negative affect	2.25	0.70	1.00–4.20
Positive affect	3.31	0.65	1.20–4.80
Shyness	2.02	1.00	1.00–5.00
Unsociability	2.36	0.91	1.00–5.00
Avoidance	1.75	0.72	1.00–4.25
Loneliness	0.88	0.70	0.00–3.00
Social anxiety	1.46	0.74	0.00–3.85
Depression	0.51	0.39	0.00–1.75
Life satisfaction	4.46	1.37	1.40–7.00
*Time 2* (*N* = 159)			
Negative affect	2.12	0.78	1.00–4.60
Positive affect	3.18	0.69	1.20–4.60
Shyness	1.67	0.74	1.00–4.25
Unsociability	2.21	0.86	1.00–4.50
Avoidance	1.61	0.64	1.00–4.25
Loneliness	0.63	0.60	0.00–2.60
Social anxiety	1.30	0.66	0.15–3.15
Depression	0.45	0.43	0.00–2.40
Life satisfaction	4.78	1.28	1.00–7.00

**Table 2 behavsci-13-00977-t002:** Zero-order correlations among study variables.

	1.	2.	3.	4.	5.	6.	7.	8.	9.	10.	11.	12.	13.	14.	15.	16.	17.	18.	19.
*Time 1*																			
1. Negative affect	–																		
2. Positive affect	−0.23 **	–																	
3. Shyness	0.44 **	−0.32 **	–																
4. Unsociability	0.09	−0.12 *	0.24 **	–															
5. Avoidance	0.28 **	−0.20 **	0.57 **	0.57 **	–														
6. Loneliness	0.47 **	−0.32 **	0.63 **	0.23 **	0.42 **	–													
7. Social anxiety	0.48 **	−0.42 **	0.79 **	0.26 **	0.52 **	0.61 **	–												
8. Depression	0.65 **	−0.34 **	0.40 **	0.10	0.30 **	0.54 **	0.39 **	–											
9. Life satisfaction	−0.41 **	0.47 **	−0.37 **	−0.18 **	−0.29 **	−0.45 **	−0.37 **	−0.54 **	–										
*Time 2*																			
10. Negative affect	0.61 **	−0.03	0.38 **	0.04	0.32 **	0.40 **	0.42 **	0.48 **	−0.26 **	–									
11. Positive affect	−0.22 **	0.65 **	−0.21 **	0.01	−0.21 **	−0.31 **	−0.35 **	−0.37 **	0.39 **	−0.16 *	–								
12. Shyness	0.35 **	−0.32 **	0.76 **	0.17 *	0.51 **	0.33 **	0.66 **	0.30 **	−0.31 **	0.42 **	−0.26 **	–							
13. Unsociability	0.04	−0.07	0.19 *	0.52 **	0.48 **	0.11	0.22 **	0.05	−0.16 *	0.15	0.01	0.24 **	–						
14. Avoidance	0.19 *	−0.29 **	0.38 **	0.35 **	0.72 **	0.43 **	0.38 **	0.19 *	−0.28 *	0.33 **	−0.21 **	0.55 **	0.41 **	–					
15. Loneliness	0.30 **	−0.23 **	0.39 **	0.09	0.32 **	0.76 **	0.39 **	0.48 **	−0.42 **	0.47 **	−0.29 **	0.46 **	0.17 *	0.45 **	–				
16. Social anxiety	0.43 **	−0.35 **	0.62 **	0.08	0.42 **	0.33 **	0.84 **	0.40 **	−0.20 *	0.51 **	−0.30 **	0.65 **	0.24 **	0.35 **	0.39 **	–			
17. Depression	0.28 **	−0.01	0.20 *	0.05	0.22 **	0.40 **	0.26 **	0.53 **	−0.27 **	0.69 **	−0.24 **	0.25 **	0.17 *	0.25 **	0.54 **	0.39 **	–		
18. Life satisfaction	−0.29 **	0.36 **	−0.31 **	−0.12	−0.38 **	−0.40 **	−0.29 **	−0.46 **	0.66 **	−0.37 **	0.40 **	−0.39 **	−0.14	−0.36 **	−0.49 **	−0.29 **	−0.43 **	–	
19. Gender	0.05	−0.07	−0.03	−0.01	−0.12 *	0.02	0.04	0.04	−0.03	−0.18 *	−0.11	−0.03	−0.12	−0.04	−0.14	0.03	−0.38	0.05	–
20. Age	−0.10	0.19 **	−0.15 **	0.05	−0.05	−0.14 **	−0.15 **	−0.09	0.07	−0.13	0.31 **	−0.08	0.09	0.03	−0.12	−0.16 *	−0.13	0.11	0.01

Note: * *p* < 0.05, ** *p* < 0.01; Gender (0 = *males*, 1 = *females*).

**Table 3 behavsci-13-00977-t003:** Model fit indices for each LPA solution.

Model	Log-Likelihood	AIC	BIC	SABIC	Entropy	Smallest Class %	LMR *p*-Value	BLRT *p*-Value
1-group	−2466.303	4952.606	4991.128	4959.405	---	---	---	---
2-groups	−2300.923	4633.846	4695.482	4644.724	0.84	27	0.47	<0.001
3-groups	−2258.033	4560.066	4644.814	4575.023	0.85	9.2	0.04	0.04
**4-groups**	**−2215.169**	**4486.338**	**4594.200**	**4505.375**	**0.88**	**6.6**	**0.05**	**<0.001**
5-groups	−2195.828	4459.655	4590.630	4482.771	0.87	3.4	0.18	0.19
6-groups	−2172.698	4425.396	4579.484	4452.591	0.87	4.6	0.37	<0.001

Note: AIC = Akaike’s information criterion; BIC = Bayesian information criterion; SABIC = sample-adjusted BIC; LMR = Lo–Mendell–Ruben; BLRT = bootstrap likelihood ratio test. In bold, we report the best LPA solution.

**Table 4 behavsci-13-00977-t004:** Characteristics of the four groups from latent profile analysis.

Variable	Shy	Unsociable	Socially Avoidant	Non-Withdrawn	*F*	*η^2^_p_*
*Time 1*						
Loneliness	1.45 _a_	1.06 _b_	1.81 _a_	0.60 _c_	58.03	0.34
Social anxiety	2.18 _b_	1.63 _c_	2.64 _a_	1.12 _d_	103.09	0.47
Depression	0.71 _a_	0.63 _a_	0.86 _a_	0.41 _b_	21.13	0.16
Life satisfaction	3.83 _bc_	4.24 _ba_	3.32 _c_	4.78 _a_	15.71	0.12
*Time 2*						
Negative affect	2.71 _a_	2.57 _a_	---	1.98 _b_	10.10	0.12
Positive affect	2.74 _b_	3.03 _ba_	---	3.27 _a_	4.84	0.06
Shyness	2.89 _a_	2.30 _b_	---	1.42 _c_	66.84	0.47
Unsociability	2.50 _ab_	2.93 _a_	---	2.07 _b_	8.07	0.09
Avoidance	1.86 _b_	2.57 _a_	---	1.45 _c_	30.86	0.29
Loneliness	1.06 _a_	0.73 _ab_	---	0.55 _b_	6.18	0.07
Social anxiety	2.24 _a_	1.77 _a_	---	1.12 _b_	35.96	0.32
Depression	0.64 _a_	0.57 _a_	---	0.41 _a_	2.70	0.03
Life satisfaction	3.91 _b_	4.49 _ba_	---	4.97 _a_	6.07	0.07

Note: For T1, all the ANOVAs were significant at *p* < 0.001. For T2, all the ANOVAs were significant at *p* < 0.001, except for positive affect (*p* = 0.01), loneliness (*p* = 0.01), depression (*p* = 0.07), and life satisfaction (*p* = 0.01). Degrees of freedom were *df* = 3, 344 at T1 and *df* = 2, 153 at T2 for all variables; *η^2^_p_* indicates partial eta-squared; means with different subscripts in the same row differ at *p* < 0.05 (Bonferroni *p*-value for multiple comparisons); socially avoidant subgroup’s means at Time 2 are not reported since this subgroup was excluded from comparisons due to small sample size (*n* = 3).

## Data Availability

Data is unavailable due to privacy.

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
