# Peer review of "Motivations for Social Withdrawal, Mental Health, and Well-Being in Emerging Adulthood: A Person-Oriented Approach"

_behavsci, 2023, doi:10.3390/bs13120977_

Round 1

Reviewer 1 Report

Comments and Suggestions for Authors

This study with n = 159 students answering questions at 2 points in time and analyzing the data with a Latent Profile Analysis (LPA) identified different groups based on characteristics of the participants. While the 2 measurement points and the LPA is of high quality, this study should make better use of the longitudinal nature of the research and built on solid and more recent theory. The current version of the manuscript does not have much innovative potential and doe not add to the literature. Thus I suggest running the LPA with the full sample at T1 again, then analyze the groups over time and analyze whether these group have any advantage to the use of manifest data. For choosing an appropriate theory, please read  https://doi.org/10.1111/aphw.12420. 

Comments on the Quality of English Language

please improve the quality of scientific aspects of the paper, and revise the writing accordingly

Author Response

Reviewer 1

This study with n = 159 students answering questions at 2 points in time and analyzing the data with a Latent Profile Analysis (LPA) identified different groups based on characteristics of the participants. While the 2 measurement points and the LPA is of high quality, this study should make better use of the longitudinal nature of the research and built on solid and more recent theory. The current version of the manuscript does not have much innovative potential and doe not add to the literature. Thus I suggest running the LPA with the full sample at T1 again, then analyze the groups over time and analyze whether these group have any advantage to the use of manifest data. For choosing an appropriate theory, please read  https://doi.org/10.1111/aphw.12420. 

Response. We thank the reviewer for this comment. In the revised manuscript, we have tried to more explicitly highlight the potential contribution of this manuscript to the extant literature (see pp. 4-5).

Regarding the LPA, we conducted analyses and reported results with the total sample at T1 (N = 348) and not only with students who participated at both time points (n = 159). Then, we compared the four individuated groups cross-sectionally, using variables at T1 (N = 348 university students) and over time, using variables collected at T2 (n = 159 university students). We have clarified the explanation of these analyses (see pp. 6, 7, 8, and Table 1). As an aside, despite the small sample size (< 300; Nylund-Gibson & Choi, 2018), results from LPA including only using students with data at both  time points (n = 159) indicated that a three subgroups solution was optimal (in the LPA with four subgroups, the avoidant subgroup was composed of a small percentage of participants – 4% –  in line with the results of ANOVAs reported in the current manuscript). At present, we have not included this information in the manuscript – but we could add this in a footnote or supplemental materials if deemed important),

We thank the reviewer for the suggested special issue regarding the theoretical framework. Since the social withdrawal literature does not typically fall under the guise of a single over-arching theoretical model, we incorporated several theories to provide a frame to underlie our research questions and hypotheses. In the revised manuscript, we have tried to make this more explicit. For example, to describe the multidimensionality of social withdrawal, we focused on Asendorpf’s (1990) motivational model (see  p. 2). We also included a similar theoretical framework cited in the special issue and proposed by Elliot et al. (2006), which postulates the presence of approach and avoidance social goals that drive individuals toward positive outcomes of social relationships (e.g., deepening one’s relationships or being accepted) or away from negative outcomes of social relationships (e.g., avoiding conflict in one’s relationships or rejection; see p. 2).

In addition, in reporting the importance of social relationships, we also described a theoretical model based on the special issue suggested by the reviewer. In line with Baumeister and Leary's (1995) Need to Belong model (see also Cohen et al., 2023), individuals are driven to form and maintain interpersonal relationships that promote positive affect. However, when belongingness needs are unmet, individuals report negative affect such as feelings of anxiety, depression, and/or loneliness.

Comments on the Quality of English Language

please improve the quality of scientific aspects of the paper, and revise the writing accordingly

Response. We have given the manuscript a careful edit in this regard.

Reviewer 2 Report

Comments and Suggestions for Authors

The current manuscript provides valuable insights into the motivations for solitude among emerging adults. While the study contributes significantly to our understanding of social withdrawal in this age group, it would benefit from the inclusion of gender and cultural context to paint a more comprehensive picture of the phenomenon.

1. Addressing Gender Differences: One critical aspect that the study lacks is an examination of potential gender differences in the reported motivations for solitude and their impact on internalizing difficulties and life satisfaction. It is well-established that gender can influence how individuals perceive and respond to social situations. Therefore, incorporating gender as a variable in the analysis could reveal whether certain motivations for solitude are more prevalent among males or females and how these motivations might relate to mental well-being and life satisfaction differently for each gender.

2. Cultural Considerations: The paper focuses solely on university students from Italy, which might limit the generalizability of the findings to other cultural contexts. Social withdrawal and its consequences can be influenced by cultural norms, societal expectations, and interpersonal dynamics. To enhance the applicability of the study's findings to a broader audience, it is essential to replicate the research across different cultural backgrounds and compare the results. This could reveal whether the motivations for solitude and their associations with internalizing difficulties and life satisfaction are consistent across cultures or if there are culture-specific patterns.

3. Intersectionality: Incorporating an intersectional approach in the study would further enrich the analysis. Intersectionality acknowledges that individuals' experiences are shaped not only by gender and cultural factors but also by the interplay of multiple identities, such as race, ethnicity, socioeconomic status, and sexual orientation. Including these additional factors in the investigation of social withdrawal motivations and their outcomes would provide a more nuanced understanding of how different dimensions of identity intersect and influence individuals' experiences of solitude and psychological well-being.

4. Qualitative Insights: While the study employs a quantitative approach using online questionnaires, integrating qualitative methods could offer a deeper understanding of the participants' lived experiences. Conducting interviews or focus groups with a diverse sample of emerging adults could reveal more personal and contextual factors influencing their motivations for solitude, thereby adding richness and depth to the findings. Also, I suggest incorporating a critical approach to understand better this aspect. An example of critical approaches to the psychology of emotions is contained in this text: Belli, S. (2023). Critical Approaches to the Psychology of Emotion. Taylor & Francis.

5. Recommendations for Support and Intervention: Lastly, considering the potential gender and cultural differences in the motivations for solitude and their outcomes, the study could conclude with practical recommendations for support and intervention tailored to specific subgroups. This could assist educators, mental health practitioners, and policymakers in developing targeted strategies to promote well-being and address potential challenges faced by socially withdrawn emerging adults.

By incorporating these suggestions, the study would significantly enhance its contribution to the literature on social withdrawal motivations among emerging adults, making it more inclusive and relevant to diverse populations.

Author Response

Reviewer 2

The current manuscript provides valuable insights into the motivations for solitude among emerging adults. While the study contributes significantly to our understanding of social withdrawal in this age group, it would benefit from the inclusion of gender and cultural context to paint a more comprehensive picture of the phenomenon.

  1. Addressing Gender Differences: One critical aspect that the study lacks is an examination of potential gender differences in the reported motivations for solitude and their impact on internalizing difficulties and life satisfaction. It is well-established that gender can influence how individuals perceive and respond to social situations. Therefore, incorporating gender as a variable in the analysis could reveal whether certain motivations for solitude are more prevalent among males or females and how these motivations might relate to mental well-being and life satisfaction differently for each gender.

Response. We thank the reviewer for this feedback, and we agree on the relevance of including gender in our analysis. However, the gender distribution of our sample (i.e., relative low proportion of males) makes the examination of gender differences difficult – particularly among the identified smaller subgroups. Of note, for socially withdrawn motivations, results from Fisher’s exact test indicated significant gender differences in the composition of the subgroups (p = .01). Results from follow-up tests (standardized residuals) indicated that the unsociable subgroup was comprised of a higher proportion of males than expected (n = 10 males vs. n = 31 females). No other differences between the observed and the expected frequencies were observed in the shy (n = 10 males vs. n = 49 females), socially avoidant (n = 1 males vs. n = 22 females), and sociable (n = 15 males vs. n = 210 females) subgroups. Regarding how these motivations might relate to mental well-being and life satisfaction differently for each gender, the small number of males in each individuated subgroup did not allow us to test the possible moderating role of gender. As an exploratory analysis, despite the small number of males in each subgroup, we tested the moderating role of gender, computing three dummy variables (i.e., shy vs other subgroups, shy and unsociable versus other subgroups, socially withdrawn subgroups versus non-withdrawn subgroup). However, results did not support any significant interaction effect between dummy variables and gender in the association with outcome variables (internalizing difficulties and life satisfaction). Therefore, we did not include this analysis in the revised manuscript for the small number of males. We included this point as a study limitation (see p. 13).

  1. Cultural Considerations: The paper focuses solely on university students from Italy, which might limit the generalizability of the findings to other cultural contexts. Social withdrawal and its consequences can be influenced by cultural norms, societal expectations, and interpersonal dynamics. To enhance the applicability of the study's findings to a broader audience, it is essential to replicate the research across different cultural backgrounds and compare the results. This could reveal whether the motivations for solitude and their associations with internalizing difficulties and life satisfaction are consistent across cultures or if there are culture-specific patterns.

Response. We thank the reviewer for this suggestion. We added this point in the future directions section (see p. 14).

  1. Intersectionality: Incorporating an intersectional approach in the study would further enrich the analysis. Intersectionality acknowledges that individuals’ experiences are shaped not only by gender and cultural factors but also by the interplay of multiple identities, such as race, ethnicity, socioeconomic status, and sexual orientation. Including these additional factors in the investigation of social withdrawal motivations and their outcomes would provide a more nuanced understanding of how different dimensions of identity intersect and influence individuals’ experiences of solitude and psychological well-being.

Response. Thanks to the reviewer for highlight this important point. Unfortunately, we did not collect other information on participants’ individual characteristics (e.g., socio-economic status and sexual orientation), but only age, ethnicity, and gender. The low percentages of males and students from other ethnicities beyond Caucasian did not allow us to test how different dimensions of identity intersect and influence individuals' experiences of solitude and psychological well-being. We added this important point in the limitations of this study (see Cheon et al., 2020 for a similar approach; see p. 13).

  1. Qualitative Insights: While the study employs a quantitative approach using online questionnaires, integrating qualitative methods could offer a deeper understanding of the participants' lived experiences. Conducting interviews or focus groups with a diverse sample of emerging adults could reveal more personal and contextual factors influencing their motivations for solitude, thereby adding richness and depth to the findings. Also, I suggest incorporating a critical approach to understand better this aspect. An example of critical approaches to the psychology of emotions is contained in this text: Belli, S. (2023). Critical Approaches to the Psychology of Emotion. Taylor & Francis.

Response. We thank the reviewer for these helpful suggestions. We added this point in the future directions.

  1. Recommendations for Support and Intervention: Lastly, considering the potential gender and cultural differences in the motivations for solitude and their outcomes, the study could conclude with practical recommendations for support and intervention tailored to specific subgroups. This could assist educators, mental health practitioners, and policymakers in developing targeted strategies to promote well-being and address potential challenges faced by socially withdrawn emerging adults.

Response. We revised the study's implications accordingly (please see the conclusion section on page 14).

Reviewer 3 Report

Comments and Suggestions for Authors

The manuscript reported findings of a latent profile analysis (LPA) aiming to identify heterogeneous patterns (or profiles) of co-occurrence of motivations for social withdrawal (i.e. shyness, unsociability and social avoidance) and affect (positive and negative) in a sample of university students. The associations of these profiles with mental health outcomes (e.g. loneliness and social anxiety) at baseline and at three-month later were examined. Four distinct profiles were identified, with each associated differentially with mental health outcomes at baseline and three-month later. The findings offer insights into distinct patterns of motivations for social withdrawal in emerging adulthood, with differential mental outcomes. The Introduction was comprehensive, and the analyses were well-executed. 

My main concern for this manuscript is the rationale to consider affect with motivations for social withdrawal simultaneously as indicator variables in the LPA, with which the identification of patterns of their co-occurrence was conducted. The Introduction was self-explanatory enough to justify the need for the identification of subgroups based on motivations for social withdrawal alone, and a LAP with motivations would have served for this purpose very well. The role of affect in LPA, both from the point of view of theory and analysis, was dubious. Results-wise speaking, the identified subgroups were mainly differentiated by distinct patterns of co-occurrence of motivations, but not affect (ref Figure 1), meaning that the inclusion of affect in the LPA might be redundant. The results may explain that labels of these profiles only reflected motivations, but not affect. Indeed, it would make more sense to consider affect as one of the mental health outcomes for comparisons between profiles. If the authors consider affect to be important in the understanding of patterns of motivations, a strong justification, both theoretically and analytically, should be made.

My other comments and suggestions are as follows:

Introduction

-        The phrase “a person-oriented approach” appeared in the title and several times throughout the manuscript. I guessed the authors wanted to contrast the LAP approach with the “variable-oriented approaches”, such as correlation and regression. The meaning of this phrase does not seem explicit. Some elaborations, perhaps in the introduction when it first appeared, would be helpful to the readers’ understanding. 

-        The rationale for the inclusion of the 3-month follow-up data was not fleshed out clearly in the Introduction. Was the 3-month follow-up analysis a proof-of-concept investigation of the comparison of mental health outcomes at baseline by profiles? Were there hypotheses concerning the development of mental health outcomes at 3-month by profiles? The Introduction should be beefed up with these arguments, as well as hypotheses, for including the 3-month follow-up analysis.

Methods:

-        The conclusion of the number of profiles and the groups of individuals in profiles are sensitive to the specification of the LPA model. Please offer details of the model specification regarding the decisions on the variance-covariance matrix of the profiles (e.g. was the covariance matrix of indicator variables set to be heterogeneous across profiles?). Please also offer details of the model optimization (e.g. number of random starts used).

Results:

-        In reporting the distribution of gender across profiles (lines 310-318, p. 9), the proportion should be mentioned in addition to the raw numbers for the readers’ easy reference.

It is a pity that there was no longitudinal analysis for the “socially avoidant” profile, due to the significant drop-out in that particular subgroup of participants (baseline: n = 23, follow-up: n = 3). Do the authors have any speculations about the high drop-out rate for that particular profile?

Author Response

Reviewer 3

My main concern for this manuscript is the rationale to consider affect with motivations for social withdrawal simultaneously as indicator variables in the LPA, with which the identification of patterns of their co-occurrence was conducted. The Introduction was self-explanatory enough to justify the need for the identification of subgroups based on motivations for social withdrawal alone, and a LAP with motivations would have served for this purpose very well. The role of affect in LPA, both from the point of view of theory and analysis, was dubious. Results-wise speaking, the identified subgroups were mainly differentiated by distinct patterns of co-occurrence of motivations, but not affect (ref Figure 1), meaning that the inclusion of affect in the LPA might be redundant. The results may explain that labels of these profiles only reflected motivations, but not affect. Indeed, it would make more sense to consider affect as one of the mental health outcomes for comparisons between profiles. If the authors consider affect to be important in the understanding of patterns of motivations, a strong justification, both theoretically and analytically, should be made.

Response. We thank the reviewer for this comment. Although Asendorpf’s (1990) theoretical model conceptualizes social withdrawal subtypes in terms of the interplay between social approach (i.e., the desire to interact with others) and avoidance (i.e., the tendency to refrain from social interactions and avoid others) motivations, contemporary perspectives of social withdrawal incorporate for motivational and affective qualities (Coplan & Bowker, 2021; Rubin et al., 2009). A similar theoretical framework has been proposed by Elliot et al. (2006; see also Nikitin & Schoch, 2021), which postulate the presence of approach and avoidance social goals that drive individuals toward positive outcomes of social relationships (e.g., deepening one’s relationships or being accepted) or away from negative outcomes of social relationships for fear of receiving negative evaluations from others (e.g., avoiding conflict in one’s relationships or rejection). In this model, social avoidance motivations are not considered as the avoidance of social interactions per se but the avoidance of threats in social situations. Overall, avoidance motivations are generally associated with negative affect (e.g., nervousness), whereas approach motivations are related to positive affect (e.g., happiness).

In addition, although we only found one subgroup of shy emerging adults, we included positive and negative affect since the presence of a positive and negative shyness has been suggested. Indeed, not all shy individuals are at risk for poorer adjustment. The positive affect experienced during social situations may yield a positive subtype of shyness, allowing these individuals to regulate their arousal and engagement with others (Asendorpf, 1990; Poole & Schmidt, 2020; see p. 3). Therefore, we preferred also to include positive and negative affect in our LPA model. Finally, the inclusion of affect in the model also allowed for more direct comparisons with previous studies taking this approach (e.g., Coplan et al., 2021).

Although no statistically significant differences emerged among the three socially withdrawn subgroups, the unsociable subgroup reported a slightly lower mean of negative affect than the other socially withdrawn subgroups and a barely higher mean of positive affect than the other socially withdrawn subgroups. In addition, results revealed that the non-socially withdrawn subgroup was statistically different from the other socially withdrawn subgroups on positive and negative affect. Finally, although there is no consensus on the number of indicators to include in a LPA model, researchers reported that more indicators lead to better findings (Wurpts & Geiser, 2014). Therefore, we preferred to include five indicator variables (including positive and negative affect) rather than three indicator variables.

Introduction

-        The phrase “a person-oriented approach” appeared in the title and several times throughout the manuscript. I guessed the authors wanted to contrast the LAP approach with the “variable-oriented approaches”, such as correlation and regression. The meaning of this phrase does not seem explicit. Some elaborations, perhaps in the introduction when it first appeared, would be helpful to the readers’ understanding. 

Response. We agree with the reviewer and we added the meaning of “a person oriented approach”  and of “a variable-oriented approach” when they appeared for the first time in the manuscript (please see on page 2).

-        The rationale for the inclusion of the 3-month follow-up data was not fleshed out clearly in the Introduction. Was the 3-month follow-up analysis a proof-of-concept investigation of the comparison of mental health outcomes at baseline by profiles? Were there hypotheses concerning the development of mental health outcomes at 3-month by profiles? The Introduction should be beefed up with these arguments, as well as hypotheses, for including the 3-month follow-up analysis.

Response. We thank the reviewer for these suggestions. We included a three-month time frame  from T1 to T2 to reflect the start and end of a semester of university (for a study aimed to explore adjustment at university using this time frame, see Gillet al., 2020). The aim was to assess the characteristics of profiles at the baseline (T1) and three-months later (when they finished the classes). We revised the Introduction and hypotheses in this revised manuscript.

Methods:

-        The conclusion of the number of profiles and the groups of individuals in profiles are sensitive to the specification of the LPA model. Please offer details of the model specification regarding the decisions on the variance-covariance matrix of the profiles (e.g. was the covariance matrix of indicator variables set to be heterogeneous across profiles?). Please also offer details of the model optimization (e.g. number of random starts used).

Response. Please find here the Mplus syntax that we used for running the LPA.

  ANALYSIS:

  Type=mixture;

  STARTS = 20 5; PROCESSORS = 1

We reported these specifications in the revised manuscript. We also increased the start values; the results were the same as those used in the analyses reported in the revised manuscript. Regarding the variance, the Mplus default is equal variances across classes (but not across variables).

Results:

-        In reporting the distribution of gender across profiles (lines 310-318, p. 9), the proportion should be mentioned in addition to the raw numbers for the readers’ easy reference.

Response. We reported the percentages, computing them on the total of males and females in each subgroup.

It is a pity that there was no longitudinal analysis for the “socially avoidant” profile, due to the significant drop-out in that particular subgroup of participants (baseline: n = 23, follow-up: n = 3). Do the authors have any speculations about the high drop-out rate for that particular profile?

Response. Unfortunately, we did not collect any other information on the drop-out (e.g., specific motivations). Still, it is possible that the high avoidance characterizing their motivation for solitude also affected their participation at T2. We included this hypothesis in the limitation of the study (on page 13).

Reviewer 4 Report

Comments and Suggestions for Authors Dear authors,
Congratulations on your excellent work presented on Motivations for Social Withdrawal, Mental Health, and Well-Being in Emerging Adulthood. As for recommendations to improve it, I recommend updating more references from the last five years on the subject of study. In addition, I suggest that you provide the DOI numbers in the References section, since although they are not mandatory, they are highly recommended to be able to cite them.

Author Response

Reviewer 4

Dear authors,
Congratulations on your excellent work presented on Motivations for Social Withdrawal, Mental Health, and Well-Being in Emerging Adulthood. As for recommendations to improve it, I recommend updating more references from the last five years on the subject of study. In addition, I suggest that you provide the DOI numbers in the References section, since although they are not mandatory, they are highly recommended to be able to cite them.

Response: We thank the reviewer for the positive comments and useful suggestions. We have updated the references cited in our manuscript with more recent studies. Due to their significance for our specific research topic, only a few studies from more than five years ago remained in the novel version of the manuscript. As requested, we have also added the DOI numbers to the Reference list.

Reviewer 5 Report

Comments and Suggestions for Authors

Thank you very much for sharing this paper with me. This is an interesting article on the topic of Motivations for Social Withdrawal, Mental Health, and Well-Being in Emerging Adulthood. The authors have explored A Person-Oriented Approach, which confirm the heterogeneity of emerging adults’ experiences of solitude. I would suggest publishing the article in present form.  

Author Response

Reviewer 5

Thank you very much for sharing this paper with me. This is an interesting article on the topic of Motivations for Social Withdrawal, Mental Health, and Well-Being in Emerging Adulthood. The authors have explored A Person-Oriented Approach, which confirm the heterogeneity of emerging adults’ experiences of solitude. I would suggest publishing the article in present form.  

Response: We kindly thank the reviewer for the appreciation.

Round 2

Reviewer 2 Report

Comments and Suggestions for Authors

I reject this scientific paper for several reasons:

1. Lack of Clarity: The paper lacks clarity in presenting its research objectives and findings. The introduction is lengthy and convoluted, making it difficult for readers to understand the main focus of the study.

2. Methodological Issues: The paper utilizes Latent Profile Analysis (LPA) to identify subgroups of socially withdrawn emerging adults, but it does not adequately explain the rationale behind this choice of analysis or how it contributes to the study's objectives. Additionally, the sample size and demographic information of participants are provided, but there is no justification for why this particular sample was chosen or how it represents the broader population.

3. Ambiguity in Terminology: The paper introduces multiple terms related to social withdrawal, such as shyness, unsociability, and social avoidance, without clear definitions or distinctions between these terms. This leads to confusion and hinders the reader's understanding of the study's key concepts.

4. Lack of Practical Implications: While the paper discusses the different subtypes of social withdrawal and their potential implications, it fails to provide concrete practical implications or recommendations for addressing these issues in emerging adults. It lacks a clear "so what" factor, leaving readers uncertain about the significance of the findings.

5. Limited Empirical Support: The paper makes broad claims about the implications of different subtypes of social withdrawal in emerging adults but relies heavily on previous research conducted in childhood and adolescence. There is a lack of empirical evidence specific to the emerging adult population.

6. Organization and Flow: The paper's organization and flow are disjointed, making it challenging for readers to follow the logical progression of the study. The research questions, methods, and results are presented in a fragmented manner.

7. Repetitive Content: The paper repeats certain points and phrases throughout the text, which adds unnecessary length without providing additional clarity or insight.

Overall, this scientific paper requires significant revisions to improve its clarity, methodology, and presentation of findings. It should also strive to provide practical implications and recommendations based on robust empirical evidence specific to the emerging adult population.

Comments on the Quality of English Language

Extensive editing of English language required

Author Response

Reviewer 2

I reject this scientific paper for several reasons:

  1. Lack of Clarity: The paper lacks clarity in presenting its research objectives and findings. The introduction is lengthy and convoluted, making it difficult for readers to understand the main focus of the study.

Response: We have better specified the main focus of the study in the Introduction section (please, see p. 5). In particular, before going into detail about the specific goals of our study, we have stated that “aimed to identify subgroups of emerging adults based on their social withdrawal motivations and positive/negative affect. To accomplish this purpose, we adopted a person-oriented approach and employed latent profile analysis in a longitudinal sample of emerging adults.”.

  1. Methodological Issues: The paper utilizes Latent Profile Analysis (LPA) to identify subgroups of socially withdrawn emerging adults, but it does not adequately explain the rationale behind this choice of analysis or how it contributes to the study's objectives. Additionally, the sample size and demographic information of participants are provided, but there is no justification for why this particular sample was chosen or how it represents the broader population.

Response: We thank the reviewer for highlighting this aspect. We have added some new content to the Introduction in order to better explain the advantages of adopting a person-oriented approach (please see p. 2). Specifically, we have specified that: “More specifically, variable-oriented approaches typically assume a homogeneous population, implying that associations between variables are uniform across all individuals. This may obscure significant individual distinctions [18]. Importantly, emerging adulthood is a unique period characterized by significant developmental changes [1]. From this perspective, a person-oriented approach can help disentangle the intricate connections between various variables by uncovering the underlying structure of the data and providing insights into how social withdrawal motivations might differentiate subgroups of emerging adults”.

 Finally, we linked this approach with the study aims.

Regarding why this sample was chosen or how it represents the broader populations, this was a convenience sample of university students. As such, we do not make any claims as to the representativeness of this sample with regard to the broader population. Notwithstanding, there are no strong theoretical reasoning why the associations among these variables might be expected to be unique among this specific sub-sample of the population. Nevertheless, we added this as a limitation of the study (on page 15).

  1. Ambiguity in Terminology: The paper introduces multiple terms related to social withdrawal, such as shyness, unsociability, and social avoidance, without clear definitions or distinctions between these terms. This leads to confusion and hinders the reader's understanding of the study's key concepts.

Response: In the revised version of the manuscript, we have clarified that “The prevailing conceptual models guiding contemporary research on social withdrawal identify and describe three distinct subtypes: shyness, unsociability, and social avoidance” (p. 2). After this initial part, there follows a detailed description of the three subtypes of social withdrawal. To facilitate the reading, we explained these terms separately in three different paragraphs, the first on shyness, the second on unsociability, and the third on social avoidance (pp. 2-3). Moreover, the distinction among these three subtypes emerges clearly from the definitions we gave. For instance, see “shyness characterizes individuals with ambivalent dispositions to interact with others and seek solitude simultaneously, reflecting an internal conflict between high social approach and avoidance motivations” (p. 2); “The second subtype of social withdrawal is unsociability, which reflects low approach and avoidance motivations to interact with others” (p. 3); and “avoidant emerging adults are believed to be not only disinterested in social interactions (in contrast to their shy counterparts), but also prone to actively avoiding opportunities for peer interaction (in contrast to their unsociable counterparts) (p. 3)”.

  1. Lack of Practical Implications: While the paper discusses the different subtypes of social withdrawal and their potential implications, it fails to provide concrete practical implications or recommendations for addressing these issues in emerging adults. It lacks a clear "so what" factor, leaving readers uncertain about the significance of the findings.

Response: We have better explained the practical implications in the Conclusions section (p. 15). In particular, we have added a part stating that “Enhanced psychological support programs could be implemented for shy and socially avoidant emerging adults to address issues related to anxiety and depression during a period of life marked by relevant challenges, such as the transition to the university environment. For instance, specific training programs aiming at enhancing emotional and social skills could benefit shy and socially anxious people [63]”.

  1. Limited Empirical Support: The paper makes broad claims about the implications of different subtypes of social withdrawal in emerging adults but relies heavily on previous research conducted in childhood and adolescence. There is a lack of empirical evidence specific to the emerging adult population.

Response: In the revised version of the manuscript, most of the cited studies strongly support our study objectives and hypotheses, drawing from the emerging adulthood literature on social withdrawal motivations. All citations reference the developmental age discussed in the text, including earlier stages of development only when relevant (e.g., childhood and adolescence). We hope that the changes made to the manuscript can be considered satisfactory.

  1. Organization and Flow: The paper's organization and flow are disjointed, making it challenging for readers to follow the logical progression of the study. The research questions, methods, and results are presented in a fragmented manner.

Response: We have re-structured the Methods section (pp. 5-7) in accord with the Action Editor’s request. We hope that these changes may have improved the organization of the parts that appeared fragmented, and that there is now greater clarity in the manuscript’s structure.

  1. Repetitive Content: The paper repeats certain points and phrases throughout the text, which adds unnecessary length without providing additional clarity or insight.

Response: We have checked the manuscript and deleted potential redundancies.

Overall, this scientific paper requires significant revisions to improve its clarity, methodology, and presentation of findings. It should also strive to provide practical implications and recommendations based on robust empirical evidence specific to the emerging adult population.

Response. We revised our manuscript following these suggestions.

Comments on the Quality of English Language

Extensive editing of English language required

Response: The manuscript has been carefully proofread and edited by one of the native English-speaking authors.

Reviewer 3 Report

Comments and Suggestions for Authors

I thank the authors for responding thoroughly to my comments. I am satisfied that my concerns have been addressed. I have no further suggestions.

Author Response

Reviewer 3

I thank the authors for responding thoroughly to my comments. I am satisfied that my concerns have been addressed. I have no further suggestions.

Response: We thank the reviewer for the constructive suggestions, which have much improved our work.

Round 3

Reviewer 2 Report

Comments and Suggestions for Authors

I am pleased to acknowledge the significant improvements made in the latest version of the manuscript. The revisions and refinements have notably enhanced the overall quality and clarity of the content.

One notable improvement is the enhanced organization of ideas, which now flows more smoothly, making it easier for readers to follow the narrative. The logical progression of concepts has added a layer of coherence to the manuscript, contributing to a more engaging reading experience.

Additionally, the incorporation of relevant literature and references has strengthened the scholarly foundation of the manuscript. The increased depth of research is evident, providing readers with a more comprehensive understanding of the subject matter. I suggest revising this new reference on well-being and quality of life in adults at work: https://journals.copmadrid.org/jwop/art/jwop2023a14 

Author Response

We are grateful that the reviewer has acknowledged the significant improvements to the manuscript. The reviewer made a final suggestion for adding a new citation to the manuscript. The suggested citation focuses on links between aspects of the working environment and well-being in the context of teleworking. This is certainly an interesting paper. However, we do not see a direct link to our study and have thus elected not to add it to the manuscript.